# The Gluten Free Diet’s Impact on Growth in Children with Celiac Disease in Two Different Countries

**DOI:** 10.3390/nu12061547

**Published:** 2020-05-26

**Authors:** Naire Sansotta, Stefano Guandalini, Simone Romano, Karine Amirikian, Marco Cipolli, Gloria Tridello, Silvia Barzaghi, Hilary Jericho

**Affiliations:** 1Department of Pediatric Hepatology, Gastroenterology and Transplantation, Hospital Papa Giovanni XXIII, 24127 Bergamo, Italy; nsansotta@asst-pg23.it; 2Department of Pediatrics, Division of Gastroenterology, Hepatology, Nutrition, Celiac Disease Center, University of Chicago Medicine, Comer Children’s Hospital, Chicago, IL 60637, USA; karine.amirikian@gmail.com (K.A.); hjericho@peds.bsd.uchicago.edu (H.J.); 3Department of Medicine, University of Verona, 37129 Verona, Italy; simone.romano@univr.it; 4Cystic Fibrosis Center, Azienda Ospedaliera Universitaria Integrata, 37126 Verona, Italy; marco.cipolli@aovr.veneto.it (M.C.); gloria.tridello@aovr.veneto.it (G.T.); 5Pediatric Department, University of Milano-Bicocca, Fondazione MBBM, San Gerardo Hospital, 20900 Monza, Italy; silvia.barzaghi8@gmail.com

**Keywords:** celiac disease, gluten free diet, body mass index

## Abstract

The effects of gluten free diet (GFD) on body mass index (BMI) and growth parameters in pediatric patients with celiac disease (CD) and their dependence on different socio-cultural environments are poorly known. We conducted an international retrospective study on celiac patients diagnosed at the University of Verona, Italy, and at the University of Chicago, Chicago, IL, USA, as underweight. A total of 140 celiac children and 140 controls (mean age 8.4 years) were enrolled in Chicago; 125 celiac children and 125 controls (mean age 7.3 years, NS) in Verona. At time of diagnosis, Italian celiac children had a weight slightly lower (*p* = 0.060) and a BMI z-score significantly (*p* < 0.001) lower than their American counterparts. On GFD, Italian celiac children showed an increased prevalence of both underweight (19%) as well as overweight (9%), while American children showed a decrease prevalence of overweight/obese. We concluded that while the GFD had a similar impact on growth of celiac children in both countries, the BMI z-score rose more in American than in Italian celiac children. Additionally, in Italy, there was an alarming increase in the proportion of celiac children becoming underweight. We speculate that lifestyle and cultural differences may explain the observed variations.

## 1. Introduction

Celiac disease (CD), is a lifelong condition that affects the small intestine in genetically susceptible individuals [1]. In children, the symptoms attributable to the disease are highly variable and are influenced by age [2]. Growth failure in terms of length (or height) or weight may be the earliest sign of the disease reported in 14% of children at diagnosis [3,4]. However, between 11% and 13% of patients have been noted to be overweight or even obese at presentation [5,6]. Currently, the only effective treatment for CD is a strict, lifelong adherence to a GFD. This typically results in resolution of small intestinal inflammation and associated symptoms [7].

While a strict GFD can be nutritionally adequate [8], if not done properly, it can have the potential to lead to certain nutritional deficiencies [9,10,11] as well as to excess caloric intake and unintentional weight gain [12].

There is little data on BMI in pediatric CD patients following a GFD [13,14,15]. Valletta et al. found that the BMI z-score increased significantly, and the percentage of overweight subjects almost doubled in 149 CD children on the GFD in Italy [16]. Again, in another Italian study, Brambilla et al. showed that the percentage of overweight subjects increased slightly in 150 children with CD on a GFD [17], in agreement with Capriati et al. who found an increase of both overweight and obese patients after GFD in 445 biopsy-confirmed CD children [18]. On the other hand, in the United States, Reilly et al. demonstrated that the BMI of 75% of the patients with a high BMI at diagnosis decreased on a GFD in 142 CD children [13].

Considering the limited available literature in children, the present study has the potential of increasing our understanding on the prevalence of normal BMI and overweight among two different cohorts of children with CD and better characterizing the changes that occur in growth indices following long term treatment with GFD, as compared to healthy peers in a follow up period of 3 years.

## 2. Materials and Methods

We conducted a large international collaborative retrospective study to compare the impact of the gluten free diet in celiac pediatric populations of two different countries: The United States of America (USA) and Italy. We hypothesized that the greater availability of industrialized gluten free products in the US would lead to a higher increase of BMI in the American celiac children than in the Italian ones.

The study population included children with CD on the GFD who were enrolled between 2002 and 2016 and were followed for more than 6 months, as well as age and gender matched healthy controls.

*Celiac disease cases*. Before inclusion in the study, a diagnosis of CD was confirmed according to the current guidelines [1,19]. The inclusion criteria for our study were positive serology and Marsh 1-3 findings on biopsy, Tissue Transglutaminase IgA (TTG-IgA) more than 10 times the upper limit of normal with positive Endomysial antibody (EMA-IgA) with or without a biopsy, or positive serology and skin biopsy for dermatitis herpetiformis. In the case of IgA deficiency, Deamidated Gliadin Peptide (DGP) IgG was used. Exclusion criteria were underlying syndromes (Down, Turner, Ehlers–Danlos). Data regarding gender, age at the diagnosis, weight, height (before GFD and during follow up), mode of disease presentation, serological assays (TTG IgA, total IgA, EMA IgA and DGP IgA/IgG levels) were collected. Mode of presentation at the time of initial CD diagnosis was classified as gastrointestinal, extraintestinal or asymptomatic. Non-adherence to the GFD was assessed through patient self-reporting to their primary physicians and evaluation of celiac serologies. Participants were classified as “strictly adherent” if they reported strict adherence during the visits and had continued improvement of their serum TTG IgA and/or EMA antibodies. Patient reports to physicians were recorded into our celiac patient registry following their visits and analyzed retrospectively for this study. Of note, all dietitians’ notes, whenever available, were reviewed to corroborate the assessment of dietetic adherence. There was no controlled system in place to further classify the extent of the patient’s dietetic adherence.

*Matched controls* were recruited among children who did not have any siblings with CD who were seen for a well-child visit, or non-GI related complaint at the emergency room, or general pediatric clinic in Chicago, US, and Verona, Italy. These healthy children were matched for age, sex and ethnicity (white non-Hispanic).

The ethical committee of the University Hospital of Verona and the Institutional Review Board of the University of Chicago Department of Pediatrics have approved the project.

### Anthropometric Assessment

Anthropometric measurements were taken according the international recommendations, using an electronic scale for weight and a stadiometer for height. Body mass index was calculated according to the weight (kg)/height (m^2^). BMI was recorded upon GFD initiation in addition to at least 1 other time point: 6 months, 1 year, 2 years, 3 years and more than 4 years.

The height, weight and BMI of celiac patients and healthy controls were converted to age-specific percentiles and z-score or standard deviation (SD) derived from growth charts published by the Center for Disease Control and Prevention (CDC) [20] for American children and by The Society for Pediatric Endocrinology and Diabetes (SIEDP) [21] for Italian children. For children younger than 2 years, weight-for height percentiles were used instead of BMI [22].

Subjects were grouped into 3 categories according to the presenting BMI percentile, as defined by the current guidelines [20]: underweight (SD or z-score < −1.65 or < 5th percentile for age), normal BMI (SD or z-score from −1.65 to 1.02 or from 5th to 85th percentile for age) and overweight (z-score >1.02 or >85th percentile for age).

Data were analyzed during the observational period and were expressed as change in height, weight and BMI z-score over time (time points: 6 months, 1 year, 2 years, 3 years and >3 years).

## 3. Results

We conducted a review of the charts available in the electronic databases of pediatric patients followed between 2002 and 2016 at the University of Chicago Celiac Disease Center (through REDCap software) and at the Verona Hospital (through Iside software). Two hundred and sixty-five celiac children (140 from Chicago) and 265 healthy children (140 from Chicago) were identified. Demographics, gender, median age at diagnosis, adherence to the GFD and duration of clinic follow up are summarized in Table 1. No statistically significant differences were found between the four groups.

The most common referral symptoms in both groups were abdominal pain, failure to thrive, short stature and decelerated growth.

### 3.1. Growth Parameters at Diagnosis

In the Italian group, 6% of patients had an abnormally high BMI and 85% presented with a normal BMI at diagnosis. The remaining 9% were underweight at the time of diagnosis (Figure 1).

Within the US Group, 17% of patients were overweight and obese, 77% had a normal BMI and 6% were underweight (Figure 1). At the time of diagnosis, Italian celiac children were noted to be thinner (*p* = 0.060) and with lower BMI scores (*p* < 0.001) as compared with US celiac children. There was no difference in height SD (*p* = 0.700). Height and weight z-scores in Italian and American celiac children at the time of diagnosis were lower than their countrymen healthy peers, while no difference was found in BMI SDs (Table 2).

### 3.2. Growth Parameters Over Time

After initiation of the GFD, both Italian and American Celiac Children had an increase in height SD (*p* < 0.001) and weight SD (*p* < 0.010) while BMI z-score did not change (*p* = 0.133 and *p* = 0.064, respectively) (Table 3).

No statistically significant difference was found in height z-score change between Italian and US celiac children (*p* = 0.700), while BMI z-score change was significant (*p* = 0.010), and weight z-score change was near significant (*p* = 0.050). The proportion of overweight children increased from 6 to 9% in Italy while the percentage of overweight and obese children decreased from 17% to 12% after initiation of the GFD in the US (Figure 2, Figure 3).

Of note, 24 of the 125 Italian CD children (19%) were found to be underweight on GFD (BMI < 5th percentile): 13 of them presented failure to thrive at diagnosis, 10 had a weight in the 5-10th percentile and 1 had normal parameters at diagnosis (weight >10th percentile). There was no statistically significant association between weight and age (median age 7.3 years, *p* = 0.1); weight and gender (*p* = 0.1) and lastly, weight and duration of follow up (*p* = 0.2). The median time elapsed between initiation of the GFD and measurement of weight and height was 4.7 years for the Italian children vs 3.19 years for US children (NS). Twenty percent of underweight Italian celiac patients were poorly adherent to the GFD and/or had one or more comorbidities (depression, H. pylori infection). As compared to Italian celiac patients, only 4% of the US celiac were underweight on GFD and most of them presented with failure to thrive at the time of diagnosis. Again, there was no statistically significant association between weight and age (median age 8.4 years, *p* = 0.9), age and either gender either duration of follow up (*p* = 0.2, *p* = 0.7). All of them were strictly adherent to the GFD, and no comorbidities were found. Overall, the proportion of overweight celiac children on the GFD was 9% in Italy vs. 12% in the US, while the proportion of underweight celiac children was 19% and 4%, respectively.

No difference in height, weight, and BMI z-score was found over time in US healthy children (*p* = 0.9327, *p* = 0.4328, *p* = 0.7850, respectively) as well as in Italian children (*p* = 0.4740, *p* = 0.6992, *p* = 0.9088, respectively). There was also no significant difference in being overweight or obese between celiac patients and healthy peers in Italy or in the US.

## 4. Discussion

Here, we describe the varied impacts of the gluten free diet (GFD) in 2 groups of celiac disease children followed in two different countries: Italy and USA. To our knowledge, while studies have been published evaluating the change in body mass index (BMI) associated with the GFD in children with CD, no study has been published thus far evaluating all growth parameters on the GFD and between two separate countries. At diagnosis, no statistically significant differences in weight and height z-scores were found between Italian and US CD children, but BMI z-scores differed within the two countries. Eighty five percent of Italian CD children had a normal BMI at diagnosis as compared to 77% of American CD patients, in agreement with previous studies [23,24,25]. In the Italian CD group, 6% of patients were overweight/obese, and 9% were underweight at diagnosis, as compared to 17% and 6% in the US, respectively. Children with untreated CD were moderately shorter, weighed less and had a slightly lower (though not statistically significant) BMIs compared to their healthy peers in both countries. Although a GFD seems to improve the weight and body mass index of children with CD [26,27], there are few data regarding growth outcomes of children with CD with a low or elevated BMI. Recent studies in US adults with CD have shown a beneficial effect of the GFD regardless of BMI at diagnosis: obese patients lost weight, whereas underweight patients gained weight [28,29]; however, there is a trend towards the development of overweight/obesity in celiac patients who strictly comply with a GFD [12,30]. In our study, we did not find a significant change in BMI z-scores in pediatric CD groups following the GFD, though both CD populations had an increase in height SD and weight SD.

The proportion of overweight CD children on the GFD increased from 6% to 9% in Italy, similar to previous results [16], whereas in the US CD group, the percentage of overweight CD children decreased from 17% to 12% on the GFD in agreement with the literature [13]. BMI increases, whether desired or undesired after treatment of childhood CD, are probably multifactorial. Improved absorption likely plays a significant role, as suggested by substantial BMI increases among children with diarrhea in recent case reports [31,32]. The lack of palatability of some gluten free foods may induce a preference for the more hyperproteic and hyperlipidic foods leading to excessive weight gain [12,33]. Lifestyle factors, particularly dietary choices, which are different in the two countries, are likely to also play a role in the change of BMI.

On the other hand, few studies have been published rising the concern of children becoming underweight after GFD. Interestingly, we found that while only 4% of US celiacs were underweight on the GFD, 19% of CD children in Italy were underweight on the GFD. Of note, 50% of Italian CD children who were underweight on the GFD had a previously normal BMI, and roughly 20% of them were found not to be adherent to the GFD and/or had other comorbidities, which were likely the main determining factors. In the US, underweight children on a GFD were previously underweight at the time of diagnosis with no other comorbidities. The reasons for these findings are unclear. We speculated that cultural difference plays a part in the availability of gluten free products and lifestyle factors. Factors likely to play a role are, for instance: home food availability, family eating habits (together as a family vs child eating apart from parents), choices of fast food over traditionally prepared foods, frequency of eating out vs home-made meals, child eating in school cafeteria vs their own home-prepared lunch, preferring a “Mediterranean diet” vs a more “Western diet”, preferentially buying and eating manufactured vs natural Gluten-Free foods, etc.

We may further speculate that many celiac patients are inadequately educated and lean towards the availability of industry-generated gluten-free options, which result in unbalanced diets and food aversion, especially in those who presented failure to thrive at diagnosis. The evaluation of possible comorbidities and adherence to the GFD are important elements in the management of celiac children even though that explains less than 20% of underweight persistence in our population. Because of the nutritional risks associated with CD, a registered dietitian must be part of the health care team that monitors the patient’s nutritional status and compliance on a regular basis in order to avoid malnourishment or worsening of previous status of overweight/obesity.

Furthermore, no difference in height, weight and BMI z-score was found over time in US healthy children or in Italian children. There was also no statistically significant difference in being overweight or obese between celiac patients and healthy peers in Italy or the US.

Overall, this study has several strengths. To the best of our knowledge, it is the first international study where celiac children from two different countries were compared with their healthy peers. The weight and height measurements were also performed according to standard procedures (as opposed to self-reported), which makes our data very reliable. As is always the case in a retrospective study, there are implicit limitations. Although all patients underwent dietary review, and most of them had at least one consultation with an experienced dietician, detailed dietary inventories and standardized questionnaires to evaluate both the degree of strictness to the GFD and of gluten contamination were not available. We compared two different populations (Italy and USA), which are not completely comparable despite the use of adequate curves. Furthermore, our cohorts derived from two tertiary care referral centers, and as a result, our experience may differ from the types of cases and presentation of CD in other centers.

## 5. Conclusions

Pediatric literature on growth parameters of children with CD on the GFD compared with healthy controls has been limited up until now. Overall, the GFD seems to improve nutritional status in American children, decreasing the proportion of overweight and underweight celiacs. In Italy, there was a slight increase in the proportion of overweight and especially of underweight celiac children following the GFD, which may be related to differences between the two countries in availability of gluten free options and/or to culturally different approaches to the management of CD by the family of celiac children. Expert dietary counseling dedicated to managing the disease may be the most important factor in the management of CD. Future, prospective studies focusing on CD patients’ choices of processed foods versus natural gluten free options in order to identify the factors responsible for these BMI alterations are needed.

## Figures and Tables

**Figure 1 nutrients-12-01547-f001:**
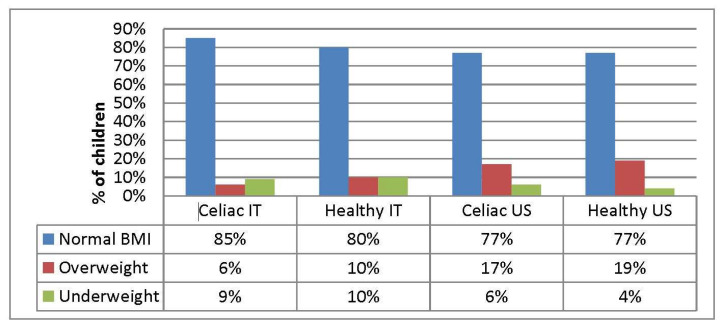
BMI at presentation.

**Figure 2 nutrients-12-01547-f002:**
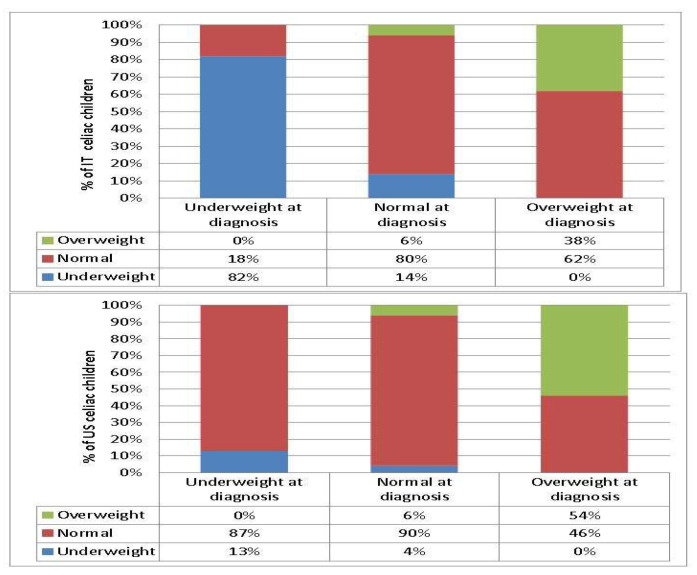
Gluten free diet (GFD) effect on BMI according to initial BMI in Italy and in the US.

**Figure 3 nutrients-12-01547-f003:**
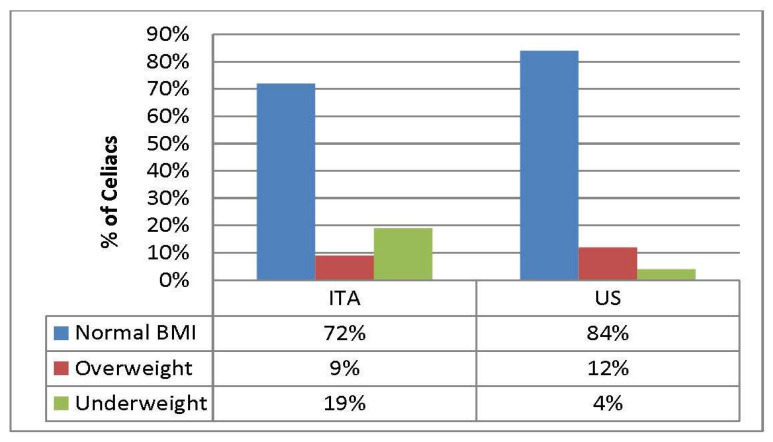
BMI at follow up in celiac groups.

**Table 1 nutrients-12-01547-t001:** Features of the enrolled patients.

	Italy		US	
	Celiac (125)	Controls (125)	Celiac (140)	Controls (140)
	N (%)	N (%)	N (%)	N (%)
Gender				
Female	93 (74.4)	83 (66.4)	93 (66.4)	90 (64.3)
Male	32 (25.6)	42 (33.6)	47 (33.6)	50 (35.7)
Age at diagnosis/time 0				
0–5.9 yrs	51 (40.8)	38 (30.4)	43 (30.7)	46 (32.8)
6–11.9 yrs	59 (47.2)	68 (54.4)	59 (42.1)	77 (55.0)
>12 yr	15 (12.0)	19 (15.2)	38 (27.2)	17 (12.2)
Median age at diagnosis/time 0	7.32	8.64	8.4	8.05
Duration of follow-up		-		
Median (years)	4.07	2.89	3.19	2.21
Range	0.53–12.58	0.52–6.75	0.51–8.77	0.59–5.17
GFD adherence (strict)	101 (80.8)	-	116 (82.8)	-

**Table 2 nutrients-12-01547-t002:** Comparison between celiac and healthy children at time 0.

Country	Italy		US	
Groups	Celiac (N = 125)	Controls (N = 125)	Celiac (N = 140)	Controls (N = 140)
**Height_SD**				
Median, range	−0.5 (−3.9–2.3)	0.3 (−2.4–2.8)	−0.5 (−4.4–2.3)	0.1 (−2.8–2.2)
Mean (SD)	−0.50 (1.09)	0.23 (0.91)	−0.59 (1.26)	0.09 (0.97)
*p* value	*p* < 0.0001 *		*p* < 0.0001 *	
**Weight_SD**				
Median, range	−0.9 (−4.1–2.5)	−0.1(−3.3–1.8)	−0.5 (−5.2–2.6)	0.1(−2.1–2.6)
Mean (SD)	−0.70 (1.09)	−0.12 (0.91)	−0.52 (1.27)	0.12 (0.94)
*p* value	*p* < 0.0001 *		*P* < 0.0001 *	
**BMI_SD**				
Median, range	−0.6 (−2.9–2.0)	−0.4 (−3.5–1.8)	0.0(−6.3–2.4)	0.1(−3.3–2.5)
Mean (SD)	−0.46 (0.99)	−0.37 (1.03)	−0.08 (1.16)	0.15 (0.98)
*p* value	*p* = 0.07		*p* = 0.44	

* statistically significant.

**Table 3 nutrients-12-01547-t003:** Changes of anthropometric parameters over time in celiac children.

	Celiac IT		Celiac US	
	Before GFD (N = 125)	After GFD (N = 125)	Before GFD (N = 140)	After GFD (N = 140)
**Height_SD**				
Median, range	−0.5(−3.9–2.3)	−0.03(−2.4–3.3)	−0.5(−4.4–2.3)	−0.24(−2.4–3.6)
Mean (SD)	−0.50 (1.09)	−0.08 (0.97)	−0.59 (1.26)	−0.2 (1.13)
*p* value	*p* < 0.0001 *		*p* < 0.0001 *	
**Weight_SD**				
Median, range	−0.9(−4.1–2.5)	−0.6(−2.7–3.03)	−0.5(−5.2–2.6)	−0.04(−2.9–3.2)
Mean (SD)	−0.70 (1.09)	−0.49 (1.07)	−0.52 (1.27)	−0.09 (1.05)
*p* value	*p* = 0.0090 *		*p* < 0.0001 *	
**BMI_SD**				
Median, range	−0.6 (−2.9–2.0)	−0.6(−3.14–2.15)	0.0(−6.3–2.4)	0.04(−2.16–2.68)
Mean (SD)	−0.46 (0.99)	−0.57 (1.11)	−0.08 (1.16)	0.07 (0.92)
*p* value	*p* = 0.1335		*p* = 0.0646	

* Statistically significant.

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
