# Peer review of "The Gluten Free Diet’s Impact on Growth in Children with Celiac Disease in Two Different Countries"

_nutrients, 2020, doi:10.3390/nu12061547_

Round 1

Reviewer 1 Report

A very interesting study in an area that is not well researched as you noted.

I do have a general comment and then a few specific notes.  The general concern is that there was no assessment of dietary adherence or diet quality of the celiac patients on the GFD.  This assessment is an important factor in the outcome of both growth and weight gain.  While your results are very interesting - without the details of the dietary intake it is hard to draw significant conclusions.

Specific notes:

Abstract line 22 and then throughout the manuscript the = sign is highlighted - why?

Line 24 - you should report the exact number or % of underweight and overweight 

page 2. line 79 - as stated above a dietary assessment or evaluation of the dietitians notes should have been reviewed for diet adherence and quality of intake

page 3. line 105 - consider rewording the sentence to be clearer

page 6. line 144 where you state there is no clear association between length of follow up and weight - again considerations of diet adherence, diet quality, the degree of natural vs processed GF products consumed  should noted especially as in line 145 - you describe that 20% of the Italian children were poorly adherent - how was this assessed?

page 7. line 187 - underweight is not only assessment of malnutrition

Author Response

A very interesting study in an area that is not well researched as you noted.

We appreciate this opening statement, thank you.

I do have a general comment and then a few specific notes.  The general concern is that there was no assessment of dietary adherence or diet quality of the celiac patients on the GFD.  This assessment is an important factor in the outcome of both growth and weight gain.  While your results are very interesting - without the details of the dietary intake it is hard to draw significant conclusions.

Thank you for this comment. Indeed, we did acknowledge this in the limitations of the study. This is now made even clearer in lines 220-223: Although all patients underwent dietary review and most of them had at least one consultation with an experienced dietician, detailed dietary inventories and standardized questionnaires to evaluate both the degree of strictness to the GFD and for gluten contamination were not available Unfortunately in fact, we could not use standardized questionnaires to evaluate both the degree of strictness to the GFD and for gluten contamination given the retrospective nature of the study.  As for assessing the adherence to the GFD (an issue well known to be difficult and prone to errors), in lines 74–76 we define as “strictly adherent”  the patients who “reported strict adherence during the visits and had continued improvement of their serum TTG IgA and/or EMA antibodies”.

Specific notes:

Abstract line 22 and then throughout the manuscript the = sign is highlighted - why?

The highlighting of the “=” is not from the authors; it must have been by the publisher, for reasons unknown to us.

Line 24 - you should report the exact number or % of underweight and overweight 

Correct. We have now done this as requested.

page 2. line 79 - as stated above a dietary assessment or evaluation of the dietitians notes should have been reviewed for diet adherence and quality of intake

Again, we acknowledge this limitation, which in a retrospective kind of study is practically unavoidable. However, even if not explicitly stated, dietitians’ notes were indeed reviewed in all cases whenever available, as now indicated in the text (lines 78-79)

page 3. line 105 - consider rewording the sentence to be clearer

We have rephrased the sentence as requested.

page 6. line 144 where you state there is no clear association between length of follow up and weight - again considerations of diet adherence, diet quality, the degree of natural vs processed GF products consumed  should noted especially as in line 145 - you describe that 20% of the Italian children were poorly adherent - how was this assessed?

This correct remark addresses again the acknowledged issue of lack of objective data about adherence to the GFD and food choices. As previously stated, adherence was judged as either “strict” or “poor” based on physicians’ assessment (that included patients’ history and evaluation of celiac serology) and dietitians’ notes. While we performed the analysis of data to the best of our ability, we once more acknowledge the limitations inherent to the retrospective nature of the investigation.

page 7. line 187 - underweight is not only assessment of malnutrition

The reviewer is correct. We have now corrected (lines 190-191) the text accordingly.

Reviewer 2 Report

The main objective of this study was comparison of GFD impacts in two cohorts (US and Italian) of celiac children. Most of the previous data from peer-reviewed literature suggest that adherence to GFD is indeed associated with obesity and/or increased BMI. Such outcome was not accomplished in this study (US cohort) although results with Italian cohort corroborated expectations. Authors suggest that such results are attributed simply to "cultural differences" but offer no clues nor evidence what these cultural differences might be. 

  • Moreover, there was no mechanism incorporated into the study design that would allow verification/evaluation of GFD adherence in both cohorts.
  • It would be more informative (although not required) to also include one cohort of healthy control children on GFD.
  • Table 3 and Figure 2 show different time points of GFD. However, it is not clear (especially in case of Fig 2) what time points are shown. Please re-work these results so it is clear how much time elapsed since initiation of GFD in each time point shown.

Conclusion: It is very difficult (impossible) to reconcile differences accomplished in this study by two cohorts of celiac children. At this point, data generated by Italian group are the ones consistent with literature and expected outcome. Data generated by US group offer very little evidence or hypothesis what exactly is going on. Therefore, these data (US cohort data) should be scrutinized and analyzed further. Without providing the plausible explanation (not wild guesses) the only publishable results in this manuscript are those generated by Italian group. 

Author Response

The main objective of this study was comparison of GFD impacts in two cohorts (US and Italian) of celiac children. Most of the previous data from peer-reviewed literature suggest that adherence to GFD is indeed associated with obesity and/or increased BMI. Such outcome was not accomplished in this study (US cohort) although results with Italian cohort corroborated expectations. Authors suggest that such results are attributed simply to "cultural differences" but offer no clues nor evidence what these cultural differences might be. 

While we agree with the reviewer that several previous studies (in adults more than in children) showed GFD to result in an increase in BMI, this is not always the case. In fact, in our introduction, we refer (see line 50) to a recent study by Reilly et al in 142 CD children in the US that found the BMI of 75% of the patients with a high BMI at diagnosis decreased on a GFD.

As for the remarks on the “cultural differences” that we refer to as likely cause of the discrepancies, please note that we deliberately chose this broad term, to signify the pool of different habits that lead to different lifestyle and dietetic choices. To name a few: home food availability, family eating habits (together as a family vs child eating apart from parents before their dinner time), choices of fast food over traditionally prepared foods, frequency of eating out vs home-made meals, child eating in school cafeteria vs their own home-prepared lunch, preferring a “Mediterranean diet” vs a more “Western diet”, preferentially buying and eating manufactured vs natural Gluten-Free foods, etc.  – This is now better explained in the text, lines 198-204.

Although it was not possible, in this retrospective analysis, to investigate each one of these factors, our hypothesis was that they were likely responsible for the different outcomes.

Moreover, there was no mechanism incorporated into the study design that would allow verification/evaluation of GFD adherence in both cohorts.

This is true, and we acknowledged the limitation, due obviously to the retrospective nature of our study (please see also response to analogous  critique by reviewer #1, and lines 220-223 of the revised manuscript: 220-223: Although all patients underwent dietary review and most of them had at least one consultation with an experienced dietician, detailed dietary inventories and standardized questionnaires to evaluate both the degree of strictness to the GFD and for gluten contamination were not available  

Assessing the adherence to the GFD is an issue well known to be difficult and prone to errors even in prospective studies. In our approach, we define as “strictly adherent” (lines 74–76) the patients who “reported strict adherence during the visits and had continued improvement of their serum TTG IgA and/or EMA antibodies”. Thus, to summarize, assessment of adherence to the GFD was based specifically on: 1) physician’s assessment – that included patients’ history and evaluation of celiac serology – and: 2) dietitians’ notes whenever available (See lines 78-79 of the revised manuscript where we now added this statement: “Of note, all dietitians’ notes, whenever available, were reviewed to corroborate the assessment of dietetic adherence”. While we performed the analysis of such data to the best of our ability, we again acknowledge the limitations inherent to the retrospective nature of the investigation.

It would be more informative (although not required) to also include one cohort of healthy control children on GFD.

The reviewer raises a very interesting point here. In reality, little is known on the effects of GFD in healthy subjects, let alone children; and it is definitely conceivable that a consistent intake of GF foods may per se cause nutritional effects. However, as the Reviewer understands, the selection of such subjects would have been difficult in a study of a retrospective nature, if not impossible given the difficulties in correctly identifying such a selected population.

Table 3 and Figure 2 show different time points of GFD. However, it is not clear (especially in case of Fig 2) what time points are shown. Please re-work these results so it is clear how much time elapsed since initiation of GFD in each time point shown.

We thank reviewer #2 for bringing up such a good point.

Table 3 shows – as indicated in its title – the changes of anthropometric parameters over time in celiac children.  In Table 1 we already specified that the median duration of follow-up was 4.7 years for the Italian children and 2.19 for the US population. This is now also reiterated in lines 146-148 by the following statement:

The median time elapsed between initiation of the GFD and measurement of weight and height was 4.7 years for the Italian children vs 3.19 years for US children (NS).

Figure 2 reports the initial and the latest BMI available for each patient.

Conclusion: It is very difficult (impossible) to reconcile differences accomplished in this study by two cohorts of celiac children. At this point, data generated by Italian group are the ones consistent with literature and expected outcome. Data generated by US group offer very little evidence or hypothesis what exactly is going on. Therefore, these data (US cohort data) should be scrutinized and analyzed further. Without providing the plausible explanation (not wild guesses) the only publishable results in this manuscript are those generated by Italian group. 

Respectfully, we did not set out to “accomplish” any specific aim: ours was an observational study aiming at comparing in 2 different settings the nutritional effect of GFD in celiac children. As for the critique of our data in US children not being “consistent with literature and expected outcome” and thus not “publishable”: respectfully, if only papers reporting data in perfect alignment with previous literature would be accepted for publication, there would be very little advancement in science and medicine, and we would be still following old teachings. Studies are done to verify new hypotheses, and as long as data are rigorously generated, analyzed and interpreted, they should not be refused, even if – or we dare saying especially when – they seem to challenge previous knowledge.

Round 2

Reviewer 2 Report

Author addressed concerns raised by review sufficiently.